# OpenReview forum: "Reliable and Efficient Amortized Model-based Evaluation"
_ICLR.cc/2025/Conference — Submitted to ICLR 2025_

### Official Review · Reviewer_9Mrp · 2024-10-28

**Soundness:** 2
**Presentation:** 2
**Contribution:** 2
**Rating:** 5
**Confidence:** 3

**Summary:**

This paper addresses the impracticality of current methods for evaluating generative models, as traditional approaches (e.g., classical test theory) are resource-intensive and unreliable when using subsets of benchmarks. To solve this, the authors propose leveraging Item Response Theory (IRT) , which decouples model ability estimation from test set selection, enhancing evaluation reliability and reducing query complexity by up to 82%. The main contriobutions are 3 fold: 1) A large-scale study showing that IRT is more efficient and reliable than traditional methods. 2) Amortized calibration, which predicts question difficulty from content, reducing calibration costs. 3) A conditional item generator that automates question creation based on difficulty, ensuring scalable item bank construction. In experiments, the authors demonstrate that the involvment of IRT can indeed reduce the cost of large scale evaluation.

**Strengths:**

1. The authors offer a solid theoretical analysis based on Item Response Theory (IRT), which effectively decouples the selection of test samples from the test taker's ability.
2. The applied IRT demonstrates greater robustness against test set selection in reflecting the model’s actual ability, as shown in Figure 3. This robustness becomes increasingly valuable as models grow larger and more resource-intensive in terms of inference costs.
3. The proposal for generating new questions using a Conditional Item Generator is promising, as discussed in Section 4.2. It introduces the possibility of automatically constructing new test datasets, which could streamline and enhance evaluation processes.

**Weaknesses:**

1. While using Item Response Theory (IRT) to model the evaluation process is valuable, it also introduces an additional understanding barrier for those unfamiliar with these concepts. For example, the term "item parameters" could be simplified to "difficulty," giving readers a more intuitive sense of what the modeling aims to capture.
2. One of the main claims made in the paper is the emphasis on "long-term, adaptive monitoring as models evolve, going beyond static benchmarking" (Line 157). However, the paper does not provide sufficient experimental evidence to support this claim. For example, it does not introduce new models to demonstrate adaptive monitoring and instead relies on conventional approaches like train/test splits. Additionally, the conditional item generator is only tested on a single dataset, which limits the generalizability and persuasiveness of the results.
3. Another critical issue is the practicality of the proposed system. Traditionally, NLP benchmarks included large datasets, making them easier to encode for analysis. However, recent evaluation trends are shifting toward more challenging tasks with relatively smaller datasets (e.g., using AIME data for testing). This work does not convincingly demonstrate its relevance or necessity in light of this trend.

**Questions:**

1. Why is it important to provide the dataset context for each question by prepending a brief description before embedding them using LLaMA-3 (as shown on lines 388-389) when fitting two models to predict item parameters?
2. What do the blue and yellow dashed lines represent in Figure 5?

---

> ### Author Response · Authors · 2024-11-24
> **Reply to Weakness**
>
> Dear reviewer 9Mrp,
>
> Thank you for your valuable feedback. First, we would like to address the weakness below.
>
> **Weaknesses 1:** While using Item Response Theory (IRT) to model the evaluation process is valuable, it also introduces an additional understanding barrier for those unfamiliar with these concepts. For example, the term "item parameters" could be simplified to "difficulty," giving readers a more intuitive sense of what the modeling aims to capture.
>
> **Answer 1:** We sincerely thank the reviewers for their insightful suggestions. To ensure clarity and accessibility for readers, we have incorporated the following clarification into the introduction part, where the terms “item” and “item parameter” first appear:
>
> The terms "item" and "question" are used interchangeably. In this paper, we exclusively use Rasch's model~\citep{rasch1993probabilistic}, a fundamental and straightforward approach within IRT, where the "item parameter" represents the difficulty level of a question.
>
> We greatly appreciate this feedback, as it has helped us enhance the paper's readability.
>
> **Weaknesses 2:** One of the main claims made in the paper is the emphasis on "long-term, adaptive monitoring as models evolve, going beyond static benchmarking" (Line 157). However, the paper does not provide sufficient experimental evidence to support this claim. For example, it does not introduce new models to demonstrate adaptive monitoring and instead relies on conventional approaches like train/test splits. Additionally, the conditional item generator is only tested on a single dataset, which limits the generalizability and persuasiveness of the results.
>
> **Answer 2:** We appreciate the insightful feedback. In our work, IRT inherently supports model monitoring by enabling the evaluation of new models. Specifically, evaluating new models can turn into monitoring when these new models represent different versions of the same model over time. Actually, experimental evidence for monitoring does exist in our results. For instance, we evaluated multiple versions of OpenAI's GPT-3.5 (0125, 0301, 0613, and 1106) on the AIR-Bench dataset. The results demonstrate that the IRT ability parameter on AIR-Bench changes over time: -0.63 (Jan 25, 2023), 0.79 (Mar 1, 2023), 0.99 (Jun 13, 2023), and 0.02 (Nov 6, 2023). These findings suggest that GPT-3.5 became progressively safer from January to June but exhibited a notable decline in safety performance with the November update. We hope this illustrates how IRT can reliably and efficiently monitor model performance as it evolves.
>
> Additionally, we acknowledge that the conditional item generator has been tested on a single dataset in the current version of our work, which limits the scope of generalizability. To address this, we have extended our approach by training a generator that can produce questions based on specified difficulty levels and desired datasets. Specifically, we combined the response matrices of 24 datasets into a single, big response matrix and conducted the joint calibration to develop a regression model that predicts item difficulty from the question embeddings. Note that each question in one single dataset is prefixed with a brief tag introducing the dataset itself to provide context information. Using this regression model, we trained the item generator. Preliminary results from this approach are promising, and we are working diligently to provide detailed experimental results before the rebuttal deadline.
>
> We sincerely appreciate your thoughtful feedback, which has greatly helped us refine and strengthen our work.
>
> **Weaknesses 3:** Another critical issue is the practicality of the proposed system. Traditionally, NLP benchmarks included large datasets, making them easier to encode for analysis. However, recent evaluation trends are shifting toward more challenging tasks with relatively smaller datasets (e.g., using AIME data for testing). This work does not convincingly demonstrate its relevance or necessity in light of this trend.
>
> **Answer 3:** We respectfully acknowledge the concern regarding the relevance of our approach given the trend toward smaller, more challenging datasets in recent evaluation tasks. However, to the best of our knowledge, current evaluation frameworks, including widely adopted standards, continue to rely extensively on large datasets for comprehensive analysis. Furthermore, while some very recent specialized benchmarks are small, the vast majority of benchmarks (both old and new) emphasize large sizes, e.g., for coverage. For instance,  AIR-Bench, a recent safety benchmark, includes 5,694 questions, requiring significant computational resources to ensure thorough assessment. We believe this underscores the ongoing importance of approaches that can effectively handle large-scale evaluations.

---

> > ### Author Response · Authors · 2024-11-24
> > **Reply to Questions**
> >
> > Regarding your questions, we address them below:
> >
> > **Question 1:** Why is it important to provide the dataset context for each question by prepending a brief description before embedding them using LLaMA-3 (as shown on lines 388-389) when fitting two models to predict item parameters?
> >
> > **Answer 1:** Thank you for your thoughtful observation. We would like to clarify that the context tag is added for training a global model on the combined dataset, which combines the 24 response matrices into a single big response matrix. This approach enables the global regression model to predict the item parameter from the embeddings of the 24 datasets while leveraging the context tag to ensure it has the necessary contextual information for generalization to new, unseen datasets. We sincerely appreciate your attention to this detail and hope this explanation provides clarity.
> >
> > **Question 2:** What do the blue and yellow dashed lines represent in Figure 5?
> >
> > **Answer 2:** Thank you for pointing this out, and we regret not clearly stating their meaning in the paper. To clarify, the blue and yellow dashed lines in Figure 5 represent the mean of the training and testing MSE across the 24 datasets, respectively. We appreciate your understanding and are grateful for the opportunity to address this oversight.

---

> > > ### Comment · Area_Chair_WRfZ · 2024-11-26
> > >
> > > Dear Reviewer 9Mrp, the ICLR discussion period is extended. Could you please take a look at the authors' rebuttal and other reviews, and see whether you would like to update your ratings? The authors would greatly appreciate your consideration and responses.

---

> > > ### Comment · Reviewer_9Mrp · 2024-11-26
> > >
> > > Thanks for the responses, I decide to keep my score unchanged.

---

> > > > ### Author Response · Authors · 2024-11-26
> > > > **Thank you for getting back to us, and asking for more guidance to improve the paper**
> > > >
> > > > Dear Reviewer 9Mrp,
> > > >
> > > > Thank you for taking the time to review our responses and for getting back to us. We appreciate your detailed feedback and constructive insights throughout this process. We understand and respect your decision to keep your score unchanged. However, if there are any additional areas where you feel we could improve or further clarify our work, we would be grateful for your guidance. Additionally, if there are any remaining questions or concerns we could address, we are more than happy to provide further details or explanations. We look forward to hearing more from you.

---

> > > > > ### Author Response · Authors · 2024-12-01
> > > > >
> > > > > Dear Reviewer 9Mrp,
> > > > >
> > > > > Thank you for your comments. We want to provide our latest result on the item generator regarding your following feedback:
> > > > >
> > > > > **Weaknesses 2:** The conditional item generator is only tested on a single dataset, which limits the generalizability and persuasiveness of the results.
> > > > >
> > > > > The item generator is trained on all datasets to generate questions given two inputs: dataset description and desired difficulty. It undergoes the same training procedure as AIR-Bench. We show the generated samples at [Generated Questions](https://drive.google.com/drive/folders/1Ga4yOYJ5g21rTVJt7DK9NKZKMtw8m3St?usp=sharing). These examples illustrate that the format, style, and content of the generated questions align well with those of the questions in the original datasets. We sincerely hope this addresses your concerns about the limitation in generalizability and persuasiveness of the item generator results.

---

### Official Review · Reviewer_V1AP · 2024-11-04

**Soundness:** 3
**Presentation:** 3
**Contribution:** 2
**Rating:** 6
**Confidence:** 2

**Summary:**

This paper introduce new method (amortized model-based evaluation) for evaluating generative models using Item Response Theory (IRT) framework. The main contributions include: 1). Using IRT to decouple model performance from test set selection, making evaluation more reliable 2). Propose "amortized calibration" technique to reduce cost of estimating item parameters 3). Develop LLM-based item generator to automatically create diverse test questions 4). Experiments with 24 NLP benchmarks and 180 language models show the proposed method is more efficient and reliable, the methodology uses Rasch's model for IRT implementation and can reduce required test samples by up to 82% compared to traditional methods. While the paper claims that it is model-based evolution and decoupled model performance from the test subset selection, it still needs to test models on some example

**Strengths:**

- Strong theoretical foundation using established IRT framework but applying in an interesting way for generative model evaluation
- Clear empirical validation with extensive experiments (24 datasets, 180 models)
- Technical details are well explained - for example, equations for Fisher information I(θ) = ∑pi(1-pi) and reliability metrics
- Good visualization and quantitative results (Figure 4 showing goodness of fit metrics)

**Weaknesses:**

- Some limitation in generated questions quality not fully addressed. Need more discussion about potential risks of automated question generation such as bias towards model-generated questions
- Empirical results focus mainly on English language tasks - multilingual evaluation missing
- No ablation studies comparing different IRT model choices
- Need more baseline comparisons with other evaluation methods (like Elo scores of LLM arena)
- While the paper claims that it is model-based evolution and decoupled model performance from the test subset selection, it still needs to test models on some examples, which is a major concern of mine

**Questions:**

No.

---

> ### Author Response · Authors · 2024-11-26
> **Responses for weaknesses**
>
> Dear reviewer V1AP,
>
> Thank you for your valuable feedback. We would like to address the weakness below.
>
> **Weaknesses 1:** Some limitation in generated questions quality not fully addressed. Need more discussion about potential risks of automated question generation such as bias towards model-generated questions
>
> **Answer 1:** We sincerely appreciate the reviewer’s thoughtful feedback on the quality of generated questions and the potential risks associated with automated question generation. We understand the importance of validating the quality of generated questions, especially the potential risk such as bias or harmful contents. To provide clarity, we have made samples of the generated questions available for review at [Generated Questions](https://drive.google.com/drive/folders/1Ga4yOYJ5g21rTVJt7DK9NKZKMtw8m3St?usp=sharing). These examples demonstrate that the format, style, and content of the generated questions align closely with those in the original datasets. Looking ahead, to automate the validation process in the absence of human oversight, we plan to use advanced models such as GPT-4 to verify semantic correctness before incorporating the questions into evaluation workflows. However, human reviewers remain indispensable in identifying, refining, and mitigating any biases that may arise in AI-generated content. This collaboration allows us to leverage the complementary strengths of both parties.
>
> Regarding the potential risks of automated question generation, we would like to first underscore the primary motivation for incorporating an automated question generator. It plays a supplemental role in adaptive testing, where questions are selected iteratively based on a desired difficulty level to evaluate model performance. While the primary source of questions is the original item pool, there may be instances where questions at a specific difficulty level are exhausted. In such cases, the item generator is employed to create additional questions. Importantly, this generator constitutes only a minor component of the evaluation process, with the majority of questions still derived from the original pool, thereby limiting the potential bias introduced by model-generated questions.
>
> We also acknowledge that automated question generation could serve broader purposes, such as replacing overused questions, expanding datasets, or constructing entirely new datasets. In these contexts, the potential for bias exists. However, as evidenced by modern AI benchmarks such as AIR-Bench [1], HEx-PHI [2], AdvBench [3], and SaladBench [4], the use of AI in dataset development is becoming increasingly common. A widely adopted practice is to involve human reviewers in the process to mitigate potential biases.
> We hope this addresses the reviewer’s concerns, and we are happy to further clarify or elaborate if needed.
>
> [1] Zeng, Y., Yang, Y., Zhou, A., Tan, J. Z., Tu, Y., Mai, Y., ... & Li, B. (2024). Air-bench 2024: A safety benchmark based on risk categories from regulations and policies. arXiv preprint arXiv:2407.17436.
>
> [2] Qi, X., Zeng, Y., Xie, T., Chen, P. Y., Jia, R., Mittal, P., & Henderson, P. Fine-tuning Aligned Language Models Compromises Safety, Even When Users Do Not Intend To!. In The Twelfth International Conference on Learning Representations.
>
> [3] Zou, A., Wang, Z., Kolter, J. Z., & Fredrikson, M. (2023). Universal and transferable adversarial attacks on aligned language models. arXiv preprint arXiv:2307.15043.
>
> [4] Li, L., Dong, B., Wang, R., Hu, et al. (2024). Salad-bench: A hierarchical and comprehensive safety benchmark for large language models. ACL 2024.

---

> ### Author Response · Authors · 2024-11-26
> **Responses for weaknesses (continue)**
>
> **Weaknesses 4:** Need more baseline comparisons with other evaluation methods (like Elo scores of LLM arena)
>
> **Answer 4:** Thank you for raising this insightful concern. We explored the possibility of incorporating Elo scores from the LLM arena into our comparisons but found that Elo scores in the LLM arena are not available for individual datasets. However, we would like to emphasize that our work includes extensive evaluations of models across 24 datasets using IRT, with comparisons to other evaluation methods including Classical Test Theory (CTT) scores and HELM leaderboard scores. For a specific dataset, the CTT score for a model is calculated as the average score across all its responses, while the HELM scores are crawled from the HELM leaderboard.
>
> In Appendix A, Figure 8 (second and third) illustrate the correlation between the IRT ability parameter $\theta$ obtained through traditional calibration and both the CTT and HELM scores. As shown, there is generally a strong correlation between $\theta$ and CTT/HELM scores across most datasets. Our findings underscore the robustness and reliability of the IRT-based evaluation framework, highlighting its alignment with established methodologies and its ability to provide valuable insights into model performance. We hope this clarification adequately addresses your concern regarding the breadth of our baseline comparisons.
>
> **Weaknesses 5:** While the paper claims that it is model-based evolution and decoupled model performance from the test subset selection, it still needs to test models on some examples, which is a major concern of mine
>
> **Answer 5:** Thank you for raising this important concern. We would like to clarify our approach. Evaluating the entire dataset across all models is often prohibitively expensive, which is why major leaderboards, such as HELM, typically evaluate only a subset of the dataset and report an average score. However, this practice introduces variability, as different leaderboards may sample different subsets, potentially leading to fluctuations in model rankings depending on the difficulty of the subsets. To address this issue, we employ an IRT-based method that allows for consistent estimation of model performance across different subsets. This approach retains the efficiency benefits of subset testing while ensuring the “item-invariance of examinee’s ability estimate,” a property that traditional methods, such as reporting an average score, do not offer. By decoupling model performance from the specific test subset, we are not suggesting that our approach does not make use of any test subset, but rather that the ability estimate remains invariant across different subsets of questions. For a more detailed explanation of this property, we refer readers to standard psychometric texts, such as Baker (2001), Chapter 5, page 90. We hope this clarifies our methodology and demonstrates the robustness of our approach. Thank you again for your thoughtful feedback, and please let us know if you have any further questions.
>
> [Baker 2001] Frank B Baker. The basics of item response theory. ERIC, 2001. Available online at https://eric.ed.gov/?id=ED458219

---

> ### Comment · Reviewer_V1AP · 2024-11-27
> **Feedback to Authors**
>
> Thanks for the detailed response from the authors, I have raised my scores.

---

> > ### Author Response · Authors · 2024-12-01
> >
> > **Weaknesses 2:** Empirical results focus mainly on English language tasks - multilingual evaluation missing
> >
> > **Answer 2:** Thank you for highlighting this important consideration. The IRT model studied in our work operates on binary responses, categorizing outcomes as either success (1) or failure (0). This design makes it naturally generalizable to multilingual datasets, as long as their results can be expressed in binary form, and the embedding model supports the specific language. To demonstrate this,  we have included the [ThaiExam dataset from HELM](https://crfm.stanford.edu/helm/thaiexam/latest/) as an additional dataset to our current experiments on 24 English datasets. As a result, we have revised the dataset number from 24 to 25 and carried out the analysis on the Thai dataset. We use four metrics to measure the calibration result:
> >
> > - Goodness of fit, which measures the accuracy of the IRT model on the response matrix for calibration, with a higher value indicating a better fit
> > - Spearman correlation between the IRT ability and CTT scores, which measures the alignment between the ability estimate from the IRT model and the average score, with higher values indicating better alignment
> > - Spearman correlation between IRT ability and HELM scores, which measures the alignment between the ability estimate from the IRT model and the HELM leaderboard score, with higher values indicating better alignment
> > - AUC-ROC leverages the ability estimates of test takers and the difficulty estimates of questions to calculate the probability of a correct answer for each entry in the response matrix, with the raw response matrix serving as the ground truth labels. Higher AUC-ROC indicates a better fit.
> >
> > We report the metric values and the standard deviation from bootstrapping on the Thai dataset as the following:
> > For traditional calibration:
> > - the goodness of fit is $0.80 \pm 0.00$
> > - the Spearman correlation between the IRT ability and CTT scores is $0.99 \pm 0.00$
> > - the Spearman correlation between IRT ability and HELM scores is $0.99 \pm 0.00$
> > - the AUC-ROC is 0.85 ± 0.00
> >
> > For the joint calibration where we amortize across the question dimension:
> > - the goodness of fit on the train split is 0.80 ± 0.00
> > - the Spearman correlation between the IRT ability and CTT scores on the train split is 0.99 ± 0.00
> > - the Spearman correlation between IRT ability and HELM scores on the train split is 0.99 ± 0.00
> > - the AUC-ROC on the train split is 0.85 ± 0.00
> > - the goodness of fit on the test split is 0.74 ± 0.00
> > - the Spearman correlation between the IRT ability and CTT scores on the test split is 0.99 ± 0.00
> > - the Spearman correlation between IRT ability and HELM scores on the train split is 0.99 ± 0.00
> > - the AUC-ROC on the test split is 0.72 ± 0.01
> >
> > These results demonstrate that the traditional calibration gives a good fit for the abilities of the test takers and the difficulties of the questions. Furthermore, the joint calibration with question dimension amortized closely approximates the behavior of the traditional calibration. We conclude that with the current multilingual capable LLM used for embedding, our method can be seamlessly applied to multilingual scenarios.
> >
> > We truly appreciate the reviewer’s thoughtful feedback, and we are confident that this addition will significantly enhance the overall comprehensiveness of our work.

---

> > > ### Author Response · Authors · 2024-12-01
> > >
> > > **Weaknesses 3:** No ablation studies comparing different IRT model choices
> > >
> > > **Answer 3:** Thank you for raising this insightful question regarding our choice of the Rasch model over 2PL or 3PL in our study. To address this concern, we conducted an ablation study on these three IRT model variants. First, let me introduce Rasch’s, 2PL and 3PL models:
> > >
> > > Each language model serves as a test taker whose ability is $\theta$. There is a question pool, where each question has a difficulty parameter $z$. A Bernoulli random variable $y$ indicates whether the test taker answers the question correctly, with $y = 1$ for a correct answer and $y = 0$ for an incorrect one. The probability of a correct answer is modeled by a logit function $p(y = 1 \mid z; \theta)$. For the Rasch's model, this probability is modeled by:
> > >
> > > \begin{align*}
> > >     p(y = 1 \mid z; \theta) = \sigma(\theta - z),
> > > \end{align*}
> > > where $\sigma$ is the sigmoid function.
> > >
> > > For the 2PL model, the discrimination parameter is introduced as a second characteristic of the questions, which is denoted as $d$. It controls the steepness of the curve that describes the probability of a correct answer. A higher value of $d$ means the probability of a correct answer is more sensitive to changes in the test taker's ability. The 2PL model is:
> > >
> > > \begin{align*}
> > > p(y = 1 \mid z; \theta, d) = \sigma(d(\theta - z)),
> > > \end{align*}
> > >
> > > For the 3PL model, the questions have a third characteristic: guessing, which is denoted as $g$, ranging between 0 and 1. It represents the probability of a correct answer by chance (for example, for four-choice questions, guessing should be 25%). The 3PL model is:
> > >
> > > \begin{align*}
> > > p(y = 1 \mid z; \theta, d, g) = g + (1 - g) \sigma(d(\theta - z))
> > > \end{align*}
> > >
> > > We compare the performance of these three IRT model variants using four metrics:
> > > - Goodness of fit, which measures the accuracy of the IRT model on the response matrix for calibration, with a higher value indicating a better fit
> > > - Spearman correlation between the IRT ability and CTT scores, which measures the alignment between the ability estimate from the IRT model and the average score, with higher values indicating better alignment
> > > - Spearman correlation between IRT ability and HELM scores, which measures the alignment between the ability estimate from the IRT model and the HELM leaderboard score, with higher values indicating better alignment
> > > - AUC-ROC leverages the ability estimates of test takers and the difficulty estimates of questions to calculate the probability of a correct answer for each entry in the response matrix, with the raw response matrix serving as the ground truth labels. Higher AUC-ROC indicates a better fit.
> > >
> > > In the following four figures, we show the performance of the three IRT models across all the datasets:
> > > - [Goodness of fit](https://anonymous.4open.science/r/reeval-1187/assets/23pl_ablation_GOF.png)
> > > - [Spearman correlation between the IRT ability and CTT scores](https://anonymous.4open.science/r/reeval-1187/assets/23pl_ablation_corr_ctt.png)
> > > - [Spearman correlation between IRT ability and HELM scores](https://anonymous.4open.science/r/reeval-1187/assets/23pl_ablation_corr_helm.png)
> > > - [AUC-ROC](https://anonymous.4open.science/r/reeval-1187/assets/23pl_ablation_AUC-ROC.png)
> > >
> > > We observe that 2PL and 3PL do not offer better performance in comparison to 1PL across the four metrics. We attribute this to the fact that we have a limited number of test takers in our dataset. With the small test taker size, we can only reliably estimate a few parameters, and introducing additional parameters—such as those in 2PL or 3PL—risks increasing the variance of our estimations because the small sample size increases the chance of recruiting biased samples. Generally, the more parameters to be inferred with a constant number of test takers, the more fluctuate the estimation might be. Consequently, we choose 1PL to help avoid these issues and facilitates a more generalizable estimation.
> > >
> > > Thanks again for your valuable question, which give us the opportunity to clarify the choice of the IRT model in our work.

---

### Official Review · Reviewer_gy8L · 2024-11-05

**Soundness:** 3
**Presentation:** 3
**Contribution:** 2
**Rating:** 8
**Confidence:** 3

**Summary:**

This paper proposes a framework for adopting language models into Item Response Theory evaluation. The model is used in two ways: first, to predict the item parameter, and second, to generate examples of a designated difficulty level.

**Strengths:**

* Innovative ideas of leveraging LLM to improve IRT evaluation, especially using item difficulty to guide RLHF.
* The experiments are through and careful.

**Weaknesses:**

* Some concerns about experiment setting and base model selection.

* This work mainly develops techniques to improve IRT-based evaluation. However, it doesn't use it to run systematic evaluations of current models, so readers can not cross-validate with other evaluation methodologies or their own experience.

**Questions:**

* Line 196, should it be $Y_{M+1}$ instead?

* In the subset evaluation experiment, the setting of having 50 hard subsets and 50 easy subsets feels artificially against CTT and in favor of IRT. I guess the differentiation of hard and easy is also based on IRT parameters, right? Then, although we can see IRT achieving consistent results in this setting, we can only infer that IRT can handle the differences in difficulty that it can capture. Because the IRT parameter is monotonic w.r.t., the correctness of each test taker, the notion of hard and simple will correlate with average accuracy. The fact that CTT shows a bimodal distribution is also largely expected. If we don't deliberately sample 50 hard and 50 simple but uniformly sample 100 subsets without any intervention, is the IRT evaluation still significantly better than CTT?

* Generating items and controlling their difficulty with RL is a clever idea. But, since we need to take some model to generate it, we ought to inspect if this asymmetry causes any unfair advantage or disadvantage to the models close to the item generation model. Suppose we have one item generation model trained from LLAMA3, another trained from GPT, and another trained from Mistral. Will the IRT scores calculated from these three item pools agree on which model is better? (A similar argument can be made about the embedding model)

* Related works: https://arxiv.org/pdf/2106.00840 https://aclanthology.org/D19-1434.pdf https://aclanthology.org/D18-1500.pdf

* Wait a second, where are the 180 language models you talked about in the abstract? Are you really human?

---

> ### Author Response · Authors · 2024-11-23
> **Reply to the Weakness: Clarifications on IRT-Based Evaluation**
>
> Dear reviewer gy8L,
>
> Thank you for your valuable feedback. First, we would like to address the weakness below.
>
> **Weaknesses 1:** Some concerns about experiment setting and base model selection.
>
> **Answer 1:** We will further address this in the Question part.
>
> **Weaknesses 2:** This work mainly develops techniques to improve IRT-based evaluation. However, it doesn't use it to run systematic evaluations of current models, so readers can not cross-validate with other evaluation methodologies or their own experience.
>
> **Answer 2:** Thank you for your thoughtful feedback. We would like to clarify that our work includes extensive evaluations of models across 24 datasets using IRT, with comparisons to established evaluation methodologies including Classical Test Theory (CTT) scores and HELM leaderboard scores. For a specific dataset, the CTT score for a model is calculated as the average score across all its responses, while the HELM scores are crawled from the HELM leaderboard.
>
> In Appendix A, Figures 8 (second and third) illustrate the correlation between the IRT ability parameter $\theta$ obtained through traditional calibration and both the CTT and HELM scores. As shown, there is generally a strong correlation between $\theta$ and CTT/HELM scores across most datasets. Additionally, Appendix A, Figure 10 (second and third), provides an analysis of the correlation between $\theta$ and CTT/HELM scores when using the joint calibration approach. We find that results from amortized calibration are comparable to those obtained with traditional calibration methods.
>
> We hope this explanation addresses the concern regarding the systematic evaluation of models. Our results demonstrate the robustness of the IRT-based evaluation framework and its alignment with established methodologies, providing valuable insights into model performance. Thank you again for highlighting this aspect, and we appreciate the opportunity to clarify.

---

> > ### Author Response · Authors · 2024-11-24
> > **Reply to Question**
> >
> > Regarding your questions, we address them below:
> >
> > **Question 2:** In the subset evaluation experiment, the setting of having 50 hard subsets and 50 easy subsets feels artificially against CTT and in favor of IRT. I guess the differentiation of hard and easy is also based on IRT parameters, right? Then, although we can see IRT achieving consistent results in this setting, we can only infer that IRT can handle the differences in difficulty that it can capture. Because the IRT parameter is monotonic w.r.t., the correctness of each test taker, the notion of hard and simple will correlate with average accuracy. The fact that CTT shows a bimodal distribution is also largely expected. If we don't deliberately sample 50 hard and 50 simple but uniformly sample 100 subsets without any intervention, is the IRT evaluation still significantly better than CTT?
> >
> > **Answer 2:** Thank you very much for your thoughtful and insightful comments. You are correct that the definitions of "hard" and "easy" subsets in our experiment are based on IRT parameters. We deeply appreciate your suggestion and we have conducted an additional experiment where IRT and CTT are compared on uniformly sampled subsets. The procedure for this experiment was as follows:
> >
> > For a given dataset, we randomly choose one test taker X to experiment. Our objective is to estimate the ability of test taker X on one subset and see whether the estimation can be generalized to another subset. Information about all other test takers is side information that all the estimation methods can use to assess the held-out test taker X. Next, two disjoint subsets of 50 questions are randomly sampled. The first subset is used to estimate the ability of test taker X, and the second subset is used to assess the generalizability of this estimation. We experiment with CTT and IRT as two estimation methods. We detail our procedure on the two subsets below:
> >
> > - In the first subset, the CTT score is calculated by averaging test taker X's answers across all questions in this subset, while the IRT score is estimated using MLE. CTT doesn’t have the mechanism to use the side information from other test takers. In contrast, IRT can exploit the side information through calibration on other test takers to identify the question parameter, which can then be used to estimate the ability of test taker X.
> >
> > - In the second subset, we predict the correctness of test taker X’s answers on this subset with the estimation obtained from the first subset. For CTT, the probability of a correct response is predicted by uniformly applying the CTT score to all questions in the second subset. IRT, using Rasch’s model,  predicts the probability of a correct response by calculating the difference between the IRT score and the specific difficulty of the question and applying the sigmoid function to it. Predicting the correctness of the answer is a binary classification task, we use the Area Under the Curve of the Receiver Operating Characteristic (AUC-ROC) as our evaluation metric, where the metric is between zero and one, and higher means better.
> >
> > Finally, to estimate the variability of AUC-ROC coming from the randomness of selecting test taker X and selecting subsets, we repeat our procedure 100 times with 10 test taker X and each has 10 different pairs of subsets. The mean and standard deviation of the AUC-ROC on the 24 datasets and the combined dataset are in the table below.
> >
> > We observed that the IRT-based approach consistently achieved higher AUC-ROC values compared to the CTT-based approach across all datasets, which demonstrates the robustness of IRT in predicting responses on unseen subsets. The results highlight that CTT is highly sensitive to the specific subset sampled, whereas IRT exhibits invariance to subset composition due to its modeling of both question difficulty and test-taker ability.
> >
> > One potential concern may be that IRT benefits from having access to question difficulty information, which could be perceived as giving it an advantage. However, we believe that this is a feature rather than a limitation. IRT's ability to leverage calibrated question difficulty information enhances its generalizability and predictive power, effectively utilizing prior information to improve performance on new data. This mirrors real-world scenarios, where leveraging information from pretraining is a common and valid practice. In contrast, CTT does not have an explicit model to incorporate such contextual information, leading to limited generalization to unseen test set.
> >
> > We hope that this additional experiment addresses your concerns and offers a more comprehensive comparison between CTT and IRT. Your feedback has been valuable in refining our work, and we are grateful for the opportunity to strengthen our analysis.

---

> > > ### Author Response · Authors · 2024-11-24
> > > **Reply to Question**
> > >
> > > Table: Generalizability of IRT and CTT estimate on unseen test set, measured by AUC-ROC
> > >
> > > |Dataset|CTT Mean |CTT Std | IRT Mean | IRT Std|
> > > |-|-|-|-|-|
> > > |boolq|0.51|0.07|0.81|0.07|
> > > |synthetic_reasoning|0.50|0.07|0.74|0.12|
> > > |mmlu|0.50|0.06|0.87|0.05|
> > > |wikifact|0.50|0.07|0.87|0.05|
> > > |math|0.50|0.06|0.83|0.11|
> > > |quac|0.52|0.07|0.82|0.07|
> > > |civil_comments|0.51|0.07|0.63|0.08|
> > > |babi_qa|0.52|0.07|0.83|0.05|
> > > |raft|0.50|0.07|0.79|0.06|
> > > |bbq|0.51|0.07|0.71|0.06|
> > > |lsat_qa|0.52|0.06|0.69|0.07|
> > > |commonsense|0.49|0.07|0.53|0.08|
> > > |truthful_qa|0.51|0.08|0.71|0.09|
> > > |synthetic_reasoning_natural|0.49|0.05|0.73|0.10|
> > > |entity_matching|0.52|0.08|0.67|0.10|
> > > |bold|0.51|0.06|0.75|0.10|
> > > |dyck_language_np3|0.51|0.07|0.78|0.07|
> > > |twitter_aae|0.50|0.06|0.98|0.02|
> > > |imdb|0.50|0.07|0.82|0.11|
> > > |narrative_qa|0.50|0.07|0.91|0.04|
> > > |legal_support|0.50|0.06|0.61|0.06|
> > > |entity_data_imputation|0.49|0.06|0.94|0.03|
> > > |airbench|0.50|0.07|0.85|0.05|
> > > |combined_data|0.50|0.06|0.82|0.08|

---

> > > > ### Author Response · Authors · 2024-11-24
> > > > **Reply to Question**
> > > >
> > > > **Question 1:** Line 196, should it be Y_{M+1} instead?
> > > >
> > > > **Answer 1:** Thank you very much for catching the oversight. You are absolutely correct—Y_{M+1} should indeed be used instead of Y_{M} to denote a new incoming question. We deeply appreciate your careful review, which have allowed us to address and rectify this issue in the revised version of our manuscript.
> > > >
> > > > **Question 4:** Related works: https://arxiv.org/pdf/2106.00840 https://aclanthology.org/D19-1434.pdf https://aclanthology.org/D18-1500.pdf
> > > >
> > > > **Answer 4:** We sincerely thank the reviewer for bringing these valuable references to our attention. We have thoroughly reviewed the suggested papers and incorporated their insights into the related work section of our manuscript. Below is the updated content that integrates these references:
> > > >
> > > > \cite{vania2021comparing} examines how test set difficulty influences deep learning performance, highlighting that models learn easy examples faster while struggling with harder examples, which is critical for designing more robust evaluations. \cite{lalor2019learning} extends this by demonstrating that IRT can be scaled using artificial response patterns from deep neural networks, significantly reducing the reliance on expensive human annotations. Together, these works underscore the role of difficulty-aware approaches in refining model evaluations and performance interpretations.
> > > > \cite{lalor2018understanding} develops IRT-based evaluation scales tailored to Natural Language Inference tasks, showing that difficulty-aware evaluation can lead to more nuanced insights into model capabilities.
> > > >
> > > > We greatly appreciate the opportunity to refine our work based on this feedback and are confident that these additions enhance the depth and context of our related work discussion.
> > > >
> > > > **Question 5:** Wait a second, where are the 180 language models you talked about in the abstract? Are you really human?
> > > >
> > > > **Answer 5:** We sincerely appreciate your feedback and the opportunity to clarify this point. We provide a comprehensive table of all models evaluated. This table includes four columns: model name, model size, pretrain dataset size, and Floating-point operations per second (FLOPs). The table is shown at https://anonymous.4open.science/r/reeval-1187/assets/model_list1.png, https://anonymous.4open.science/r/reeval-1187/assets/model_list2.png, and https://anonymous.4open.science/r/reeval-1187/assets/model_list3.png.
> > > >
> > > > Furthermore, we acknowledge that not every model is evaluated in every dataset. To enhance clarity and transparency, we have included a table explicitly indicating the presence of each model across the datasets. In the following figure, each row represents a dataset, and each column represents a model. A blue block indicates that a specific model is present within a dataset. The figure is shown at https://anonymous.4open.science/r/reeval-1187/assets/dataset_model_stat.png.
> > > >
> > > > Finally, we would like to report an updata: the total number of models evaluated has been revised from 180 to 161. This change resulted from a thorough review during which we identified and removed duplicate entries in our dataset (e.g., "llama3_8b" and "Llama3_8B"). We emphasize that this fix does not affect any other quantitative results or qualitative results. We sincerely apologize for the initial misreporting and have taken steps to ensure that all presented data is now accurate and consistent. We appreciate your feedback, which has allowed us to improve the precision and transparency of our work.

---

### Official Review · Reviewer_c8mZ · 2024-11-05

**Soundness:** 2
**Presentation:** 3
**Contribution:** 4
**Rating:** 6
**Confidence:** 4

**Summary:**

The paper proposes a new way of evaluating generative models that leverages Item Response Theory (IRT) to form a more cost efficient and robust evaluation framework. The authors design both an amortized calibration algorithm as well as train a conditional item generator to aid in this process. The amortized calibration reduces the cost of estimating item parameters and reduces the cost complexity from linear to constant assuming a fixed question bank size. The item generator network uses a fine-tuned language model to generate test questions which can be conditioned on difficulty level.

**Strengths:**

This work has many strengths. First, the paper presents a novel approach to evaluating generative models that addresses the high cost and potential pitfalls of  current evaluation techniques. As far as I know, this is the first work to propose a fine-tuned language model to generate novel evaluation questions. The authors present a strong knowledge of the underlying IRT theory and write the math for the method behind it clearly and concisely.

**Weaknesses:**

The most glaring weakness of the paper is that they claim to test 180 models and don’t show the results of all of them anywhere in the paper or an appendix. The closest I could see of all of the models evaluated was in Figure 1, but it is truncated to fit in the paper; a table or chart at least demonstrating the model performance on the conditionally generated dataset and comparing it to the subsets of HELM would have been useful. Some further justification for assumptions behind the method of their IRT formulation would have been useful as well. (addressed further in the questions section below)

Some formatting concerns:
 - the references aren’t linked
 - the figures aren’t linked
 - some of the graphs are hard to read when printed due to small text

**Questions:**

- How do you make sure that the amortized calibration processed doesn’t worsen the bias present in the training data?
- What motivated the choice of Rasch’s model over other IRT models like 2PL or 3PL? Was it a simplification made to make the model tractable?
- How do you validate the quality and correctness of the generated questions? Specifically, how do you deal with questions that might match the desired difficulty but be semantically invalid, without human oversight?
- Did you compare the amount of compute needed to create the amortized model with the compute needed to evaluate a single model? How many models do you need to evaluate to have the amortized model be more efficient?

---

> ### Author Response · Authors · 2024-11-25
> **Responses for weaknesses**
>
> Dear reviewer c8mZ,
>
> Thank you for your valuable feedback. First, we would like to address the weakness below.
>
> **Weaknesses 1:** The most glaring weakness of the paper is that they claim to test 180 models and don’t show the results of all of them anywhere in the paper or an appendix. The closest I could see of all of the models evaluated was in Figure 1, but it is truncated to fit in the paper
>
> **Answer 1:**
> Thank you for your detailed feedback and for highlighting this critical point. To address your concern, we have provided a comprehensive table listing all models evaluated in our study. The full table is accessible at the following links: [model list part 1](https://anonymous.4open.science/r/reeval/assets/model_list1.png), [model list part 2](https://anonymous.4open.science/r/reeval/assets/model_list2.png), and [model list part 3](https://anonymous.4open.science/r/reeval/assets/model_list3.png).
>
> To further enhance transparency, we have also included a dataset-model matrix to document the presence of models across different datasets. In this matrix, rows represent datasets, columns represent models, and blue blocks indicate that a specific model is evaluated on a given dataset.
> [dataset-model presence matrix](https://anonymous.4open.science/r/reeval/assets/dataset_model_stat.png)
>
> We want to point out that the evaluation results of these models are available on HELM. The value of our work is making LLM evaluation more efficient and reliable through the use of IRT. In this paper, we have demonstrated that the result from our method is highly agreeable with HELM as indicated in Figure 4. We appreciate your thoughtful review, which has helped us improve the clarity and transparency of our work. Thank you for providing the opportunity to address these important concerns.
>
> **Weaknesses 2:** a table or chart at least demonstrating the model performance on the conditionally generated dataset and comparing it to the subsets of HELM would have been useful.
>
> **Answer 2:** At first, we appreciate your insightful question. We are currently conducting experiments to address this and are evaluating the performance of several models on the newly generated AIR-Bench-style questions. These results will be compared with their performance on the original AIR-Bench questions to provide a clearer understanding of the differences and insights gained.
>
> **Weaknesses 3:** Some formatting concerns:
> the references aren’t linked
> the figures aren’t linked
> some of the graphs are hard to read when printed due to small text
>
> **Answer 3:** We thank you for highlighting these formatting issues. We have carefully addressed all the concerns in the revised submission. Specifically, we have ensured that references are properly linked, figures are correctly linked, and graphs are adjusted to enhance readability. We greatly appreciate your feedback, which has helped us improve the clarity and presentation of our work.

---

> ### Author Response · Authors · 2024-11-25
> **Responses for questions**
>
> Regarding your questions, we address them below:
>
> **Question 3:** How do you validate the quality and correctness of the generated questions? Specifically, how do you deal with questions that might match the desired difficulty but be semantically invalid, without human oversight?
>
> **Answer 3:** Thank you for your insightful question. We understand the importance of validating the quality and correctness of generated questions, especially in cases where they may match the desired difficulty but lack semantic validity. To provide clarity, we have made samples of the generated questions available for review at [Generated Questions](https://drive.google.com/drive/folders/1Ga4yOYJ5g21rTVJt7DK9NKZKMtw8m3St?usp=sharing). These examples demonstrate that the format, style, and content of the generated questions align closely with those in the original datasets. Additionally, we conducted human evaluations on all generated questions, confirming that they are semantically valid and align well with their intended purpose. Looking ahead, to automate the validation process in the absence of human oversight, we plan to use advanced models such as GPT-4 to verify semantic correctness before incorporating the questions into evaluation workflows. From a theoretical perspective, the embedding model used in our approach (currently Llama 3 8B) plays a crucial role in distinguishing semantically valid questions. Since semantically invalid questions would likely produce embedding vectors that deviate significantly from those of valid ones, their predicted difficulty would also diverge from the target level. Consequently, such invalid questions are less likely to be selected during the generation process, as they would be outperformed by semantically valid questions with more accurate difficulty predictions. We sincerely appreciate your thoughtful feedback, which helps us enhance the robustness and clarity of our methodology. Thank you for engaging with our work so constructively.
>
> **Question 4:** Did you compare the amount of compute needed to create the amortized model with the compute needed to evaluate a single model? How many models do you need to evaluate to have the amortized model be more efficient?
>
> **Answer 4:** Thank you for your excellent question. Our proposed framework consists of two phases: calibration and scoring. The creation of the amortized model is done in the calibration phase while evaluating a single LLM is in the second phase.
>
> Firstly, in the calibration phase, we build the amortized model using responses from dozens of LLMs, which are sourced from existing leaderboards and benchmarks. To create this model, we first compute embeddings for all available questions across our datasets (each question corresponds to a 4096-dim vector) and then optimize the parameters of the amortized model using techniques such as EM or MLE. The embedding model we use is Llama 3 8B, and the amortized model is a simple 2-layer 4096-hidden fully-connected network. This step incurs a one-time cost, which is much smaller than evaluating a single LLM of comparable size, as the cost of inferring embeddings is equivalent to generating one token using an LLM, and the cost for training the amortized model is neglectable in comparison to the LLM inference cost.
>
> Secondly, in the scoring phase, while evaluating new LLM, we need the question difficulties to infer ability. The challenge arises when the questions are new such as when we need to replenish the item bank with new questions due to contamination or the question overused. Then, if we have an amortized network obtained from phase one, the computational cost for inferring difficulty is number_of_questions x (embedding_per_question + forward pass cost for 2-layer network). In contrast, without the amortized model, estimating question difficulty would typically require running dozens of LLM models (ranging in size from small 3B to larger 70B) on those new questions to gather sufficient data for reliable estimation. We would also like to emphasize that running an LLM on a question involves multiple forward passes (equal to the number of generated tokens), and evaluating the outputs of the LLM incurs additional costs, such as querying GPT-as-a-Judge or human annotators, which is very expensive. If we are not concerned about evaluating the LLMs on new questions, we already have the question difficulty (obtained from the calibration phrase). Thus, we do not need the amortized network.
>
> Therefore, the amortized model demonstrates significant computational savings, particularly when evaluating newly added questions. In summary, while training the amortized model incurs a small upfront computational cost, this investment pays off substantially in terms of efficiency for subsequent evaluations. We hope this explanation provides clarity and effectively addresses your concerns. Thank you again for your thoughtful question—it has allowed us to highlight the practical benefits of our approach.

---

> > ### Author Response · Authors · 2024-12-01
> >
> > **Weaknesses 2:** a table or chart at least demonstrating the model performance on the conditionally generated dataset and comparing it to the subsets of HELM would have been useful.
> >
> > **Answer 2:** We are pleased to share our latest results, which compare model performance on the conditionally generated dataset and the original dataset. Here is the experiment design: we fine-tuned two item generators using the same fine-tuning procedure but with different base models—one based on Llama3 8B and the other on Mistral 7B v0.3. These models were used to generate two distinct AIR-Bench-style question pools, each containing 1,000 questions. Together with the original AIR-Bench questions, these pools were used to query 27 language models. The models' responses were then converted into binary patterns using GPT4o-as-a-judge, which outputs 0 or 1 based on the question and the model answer. Specifically, the model fails (score of 0) if it completes an unsafe request and passes (score of 1) if it declines the request.
> >
> > This process generated three response matrices—one for each question pool (original AIR-Bench, Llama3-generated questions, and Mistral-generated questions), denoted as $Y_1$, $Y_2$, and $Y_3$, respectively. These matrices were concatenated along the question dimension to form $Y_{all}$. For $Y_1$, the section corresponding to the 27 training models is denoted as $Y_{1, train}$. Similarly, $Y_{all, train}$ represents the combined training set responses.  We then performed calibration on $Y_{all, train}$ to estimate the difficulty parameters of the questions and the ability parameters of the test takers, $\theta_{all, train}$. Calibration was conducted jointly across all three question sets, as the relative nature of question difficulty necessitates normalization during calibration. Calibrating each response matrix separately would yield difficulty parameters that are not directly comparable. As a reference, we also calibrated the responses from the original AIR-Bench ($Y_{1, train}$) independently, yielding difficulty estimates and ability parameters $\theta_{org, train}$.
> >
> > For the 27 training models, the Spearman correlation between $\theta_{all, train}$ and $\theta_{org, train}$ was 0.96. Additionally, the Spearman correlation between $\theta_{all, train}$ and the classical test theory (CTT) scores from the original AIR-Bench was also 0.96. These findings demonstrate that the conditionally generated questions successfully meet the target difficulty levels and are reliable for evaluating model performance. We hope this additional experiment addresses your concerns. Thank you for your thoughtful suggestion, which has further strengthened the robustness of our work.

---

> > > ### Author Response · Authors · 2024-12-01
> > >
> > > **Question 2:** What motivated the choice of Rasch’s model over other IRT models like 2PL or 3PL? Was it a simplification made to make the model tractable?
> > >
> > > **Answer 2:** Thank you for raising this insightful question regarding our choice of the Rasch model over 2PL or 3PL in our study. To address this concern, we conducted an ablation study on these three IRT model variants. First, let me introduce Rasch’s, 2PL and 3PL models:
> > >
> > > Each language model serves as a test taker whose ability is $\theta$. There is a question pool, where each question has a difficulty parameter $z$. A Bernoulli random variable $y$ indicates whether the test taker answers the question correctly, with $y = 1$ for a correct answer and $y = 0$ for an incorrect one. The probability of a correct answer is modeled by a logit function $p(y = 1 \mid z; \theta)$. For the Rasch's model, this probability is modeled by:
> > > \begin{align*}
> > >     p(y = 1 \mid z; \theta) = \sigma(\theta - z),
> > > \end{align*}
> > > where $\sigma$ is the sigmoid function.
> > >
> > > For the 2PL model, the discrimination parameter is introduced as a second characteristic of the questions, which is denoted as $d$. It controls the steepness of the curve that describes the probability of a correct answer. A higher value of $d$ means the probability of a correct answer is more sensitive to changes in the test taker's ability. The 2PL model is:
> > >
> > > \begin{align*}
> > > p(y = 1 \mid z; \theta, d) = \sigma(d(\theta - z)),
> > > \end{align*}
> > >
> > > For the 3PL model, the questions have a third characteristic: guessing, which is denoted as $g$, ranging between 0 and 1. It represents the probability of a correct answer by chance (for example, for four-choice questions, guessing should be 25%). The 3PL model is:
> > >
> > > \begin{align*}
> > > p(y = 1 \mid z; \theta, d, g) = g + (1 - g) \sigma(d(\theta - z))
> > > \end{align*}
> > >
> > > We compare the performance of these three IRT model variants using four metrics:
> > > - Goodness of fit, which measures the accuracy of the IRT model on the response matrix for calibration, with a higher value indicating a better fit
> > > - Spearman correlation between the IRT ability and CTT scores, which measures the alignment between the ability estimate from the IRT model and the average score, with higher values indicating better alignment
> > > - Spearman correlation between IRT ability and HELM scores, which measures the alignment between the ability estimate from the IRT model and the HELM leaderboard score, with higher values indicating better alignment
> > > - AUC-ROC leverages the ability estimates of test takers and the difficulty estimates of questions to calculate the probability of a correct answer for each entry in the response matrix, with the raw response matrix serving as the ground truth labels. Higher AUC-ROC indicates a better fit.
> > >
> > > In the following four figures, we show the performance of the three IRT models across all the datasets:
> > > - [Goodness of fit](https://anonymous.4open.science/r/reeval-1187/assets/23pl_ablation_GOF.png)
> > > - [Spearman correlation between the IRT ability and CTT scores](https://anonymous.4open.science/r/reeval-1187/assets/23pl_ablation_corr_ctt.png)
> > > - [Spearman correlation between IRT ability and HELM scores](https://anonymous.4open.science/r/reeval-1187/assets/23pl_ablation_corr_helm.png)
> > > - [AUC-ROC](https://anonymous.4open.science/r/reeval-1187/assets/23pl_ablation_AUC-ROC.png)
> > >
> > > We observe that 2PL and 3PL do not offer better performance in comparison to 1PL across the four metrics. We attribute this to the fact that we have a limited number of test takers in our dataset. With the small test taker size, we can only reliably estimate a few parameters, and introducing additional parameters—such as those in 2PL or 3PL—risks increasing the variance of our estimations because the small sample size increases the chance of recruiting biased samples. Generally, the more parameters to be inferred with a constant number of test takers, the more fluctuate the estimation might be. Consequently, we choose 1PL to help avoid these issues and facilitates a more generalizable estimation.
> > >
> > > Thanks again for your valuable question, which allows us to clarify the choice of the IRT model in our work.

---

> > > > ### Author Response · Authors · 2024-12-02
> > > >
> > > > **Question 1:** How do you make sure that the amortized calibration processed doesn’t worsen the bias present in the training data?
> > > >
> > > > **Answer 1:** Thank you for highlighting the important concern of data bias. We understand this as a statement about how biases present in the training data might lead to evaluation bias. The goal of amortized calibration is to closely approximate the behavior of traditional/non-amortized calibration at a significantly lower cost. We demonstrate that amortized calibration does not introduce systematic bias in the evaluation result through four metrics—Goodness of Fit, Spearman correlation between IRT ability and CTT scores, Spearman correlation between IRT ability and HELM scores, and AUC-ROC. Experiments were conducted on all datasets with a train-test split along the question dimension. In the following figures, blue points represent the train split of each dataset, while red points represent the test split. The x-axis corresponds to the amortized calibration metric values, and the y-axis corresponds to the non-amortized calibration metric values.
> > > >
> > > > - [Goodness of fit](https://anonymous.4open.science/r/reeval-1187/assets/gof_nonamor_amor.png)
> > > > - [Spearman correlation between the IRT ability and CTT scores](https://anonymous.4open.science/r/reeval-1187/assets/ctt_nonamor_amor.png)
> > > > - [Spearman correlation between IRT ability and HELM scores](https://anonymous.4open.science/r/reeval-1187/assets/helm_nonamor_amor.png)
> > > > - [AUC-ROC](https://anonymous.4open.science/r/reeval-1187/assets/auc_nonamor_amor.png)
> > > >
> > > > These results show strong alignment between amortized and non-amortized calibration. Hence we conclude that the metric values align well with each other, demonstrating that the amortized result closely approximates that of traditional/non-amortized calibration without introducing significant bias in evaluation result.
> > > >
> > > > At the same time, we also carry out an ablation study on the embedding model for amortized calibration. Specifically, we obtain the embeddings of the questions using two different embedding models: Llama3 8B and Mistral 7B v0.3, and joint calibrations are conducted with the embedding from each embedding model with a train test split in the question dimension. We also show the correlation of the above four metrics between the results from the two embedding models. In the following figures, each blue point is the train split of a dataset, and each red point is the test split of a dataset. The x-axis value of the point represents the metric value from Llama 3 8B embedding, and the y-axis value of the point represents the metric value from Mistral 7B v0.3 embedding.
> > > >
> > > > - [Goodness of fit](https://anonymous.4open.science/r/reeval-1187/assets/emb_ablation_gof.png)
> > > > - [Spearman correlation between the IRT ability and CTT scores](https://anonymous.4open.science/r/reeval-1187/assets/emb_ablation_ctt.png)
> > > > - [Spearman correlation between IRT ability and HELM scores](https://anonymous.4open.science/r/reeval-1187/assets/emb_ablation_helm.png)
> > > > - [AUC-ROC](https://anonymous.4open.science/r/reeval-1187/assets/emb_ablation_auc.png)
> > > >
> > > > We conclude that the metric values align well with each other, and the embedding model has a minor influence on the calibration result.
> > > >
> > > > With these results, we believe that our amortized calibration approach does not introduce systematic bias in the evaluation result, in comparison to the traditional evaluatio technique. We appreciate the opportunity to address this and welcome further suggestions to refine our methodology.

---

### Official Review · Reviewer_D8pS · 2024-11-06

**Soundness:** 3
**Presentation:** 4
**Contribution:** 3
**Rating:** 8
**Confidence:** 3

**Summary:**

In this paper, the authors tackle the problem of reducing the cost of models in the context of generative models and LLMs. The authors propose to evaluate models by modeling evaluation using Item Response Theory - framing evaluation as predicting the abilities of a learner exposed to questions/items. While this can already reduce evaluation cost, the authors propose two methods to further reduce cost/improve the reliability of the evaluation (i) parameterizing evaluation questions with an item model and (ii) adaptively testing learners by generating questions from item parameters using a generative model.

**Strengths:**

This manuscript considers an important but often neglected cost for developing models - the compute required for extensive evaluation.

Drawing on item response theory is a refreshing way of reframing the problem to yield new insights.

Adaptive testing with conditional item generation is a particularly interesting idea, as the evaluation is tailored to the model. This idea also partially addresses another potential problem for model evaluation - overfitting/evaluation leakage.

**Weaknesses:**

No major weaknesses, however see my question below.

**Questions:**

Generating questions for adaptive testing is clearly useful, but what do the the generated questions look like? How does the underlying generative model affect the results? How can we automatically verify the generated questions are valid/correct?

---

> ### Author Response · Authors · 2024-12-01
>
> Dear reviewer D8pS,
>
> Thank you for your valuable feedback. We would like to address the questions below.
>
> **Question 1:** Generating questions for adaptive testing is clearly useful, but what do the the generated questions look like?
>
> **Answer 1:** Thank you for your thoughtful and insightful question. To provide further clarity, we provide generated questions from our finetuned generator(which is trained on all questions from all datasets). The generated questions are accessible at [Generated Questions](https://drive.google.com/drive/folders/1Ga4yOYJ5g21rTVJt7DK9NKZKMtw8m3St?usp=sharing). These examples illustrate that the format, style, and content of the generated questions align well with those of the questions in the original datasets. We sincerely hope this addresses your concern, and we deeply appreciate your interest in our work.
>
> **Question 2:** How does the underlying generative model affect the results?
> Thank you for your thoughtful question. To address it, we conducted an ablation study to investigate the performance of item generators based on different underlying generative models. Here is the design of our experiment: we fine-tuned two item generators using the same fine-tuning procedure but with different base models. One generator utilized Llama3 8B, while the other used Mistral 7B v0.3. Each generator produced a distinct AIR-Bench-style question pool containing 1,000 questions. Together with the original AIR-Bench questions, these pools were used to query 35 language models, comprising 27 training models and 8 testing models.
>
> The models' responses were then converted into binary patterns using GPT4o-as-a-judge, which assigned a score of 0 (failure) when the model completed an unsafe request and a score of 1 (success) when the model declined the request. This process generated three response matrices—one for each question pool (original AIR-Bench, Llama3-generated questions, and Mistral-generated questions), denoted as $Y_1$, $Y_2$, and $Y_3$, respectively. These matrices were concatenated along the question dimension to create $Y_{all}$.
> For $Y_1$, the section corresponding to the 27 training models is denoted as $Y_{1, train}$, and the section for the 8 testing models as $Y_{1, test}$. Similarly, we define $Y_{2, train}$, $Y_{2, test}$, $Y_{3, train}$, $Y_{3, test}$, $Y_{all, train}$, and $Y_{all, test}$. Subsequently, we performed calibration on $Y_{all, train}$ to estimate the difficulty parameters of the questions, $z_{all, train} = [z_{all, train, 1}, z_{all, train, 2}, z_{all, train, 3}]$, and the ability parameters of the models, $\theta_{all, train}$. This joint calibration was necessary because question difficulty is a relative attribute, making meaningful comparisons possible only when all sets are calibrated together. For reference, calibration was also performed solely on the original AIR-Bench responses ($Y_{1, train}$), producing difficulty estimates $z_{org, train} = [z_{org, train, 1}]$ and ability estimates $\theta_{org, train}$.
> For the 8 testing models, we inferred their ability parameters using the difficulty parameters and responses from the original AIR-Bench ($z_{org, train, 1}$ and $Y_{1, test}$), denoted as $\theta_{org, test, 1}$. Similarly, we inferred the ability of the testing models using $z_{all, train, 2}$ and $Y_{2, test}$, as well as $z_{all, train, 3}$ and $Y_{3, test}$, resulting in $\theta_{all, test, 2}$ and $\theta_{all, test, 3}$, respectively. The correlations between $\theta_{org, test, 1}$ and $\theta_{all, test, 2}$, as well as between $\theta_{org, test, 1}$ and $\theta_{all, test, 3}$, were both 0.81. These results demonstrate that the choice of base model for the item generator does not significantly impact the ability estimates of the testing models. We hope this study addresses your question thoroughly. Thank you for your insightful feedback, which has contributed to the robustness of our work.
>
> **Question 3:** How can we automatically verify the generated questions are valid/correct?
>
> **Answer 3:** Thank you for raising this important question. Ensuring the correctness of generated questions involves multiple dimensions, including semantic accuracy and alignment with the target difficulty. For example, in a safety dataset like AIR-Bench, it is essential that the generated questions effectively measure safety. To assess semantic correctness, we conducted human evaluations on a subset of the generated questions, confirming their validity. Moving forward, we aim to automate this validation process by leveraging advanced models such as GPT-4 to verify semantic correctness before integrating the questions into evaluation workflows. We appreciate your thoughtful feedback, which has helped us refine and clarify our methodology. Thank you again for your question.

---

### Official Review · Reviewer_tNxg · 2024-11-07

**Soundness:** 3
**Presentation:** 3
**Contribution:** 3
**Rating:** 6
**Confidence:** 3

**Summary:**

The paper proposes a novel model-based evaluation framework using Item Response Theory (IRT) to efficiently and reliably evaluate generative models. The key innovations include amortized calibration to reduce the cost of estimating item parameters and a conditional item generator to automate diverse question generation. The approach is validated on 24 benchmarks and 180 language models.

**Strengths:**

1. The application of IRT in generative model evaluation addresses the limitations of classical test theory (CTT) in machine learning.
2. Amortized calibration significantly reduces the cost and complexity of item parameter estimation, making continuous evaluation feasible.
3. The item generator automates question creation, which is crucial for maintaining a diverse and up-to-date item bank.
4. Experiments are very thorough.

**Weaknesses:**

1. The introduction of IRT and amortized calibration adds complexity to the evaluation process, which might be challenging for practitioners without a background in psychometrics.
2. Details on the practical implementation of the item generator and its integration into existing workflows are limited.
3. The integration of Item Response Theory (IRT) with machine learning evaluations assumes a level of prior knowledge (e.g., psychometric concepts) that could limit accessibility.
4. The experiments predominantly rely on synthetic setups or controlled environments. Real-world case studies or collaborations with industry practitioners would strengthen the empirical validation.

**Questions:**

1. The paper mentions missing values in the response matrix. How are these handled during calibration and scoring to ensure accurate ability estimation?
2. Is the IRT model effective for non-text data, such as images or multimodal datasets? What adjustments are necessary?
3. How does the automated question generation avoid introducing bias, ensuring fair and objective evaluations?

---

> ### Author Response · Authors · 2024-11-25
> **Responses for weaknesses**
>
> Dear Reviewer tNxg,
>
> Thank you for your valuable feedback. We answer your comment below.
>
> **Weaknesses 1:** The introduction of IRT and amortized calibration adds complexity to the evaluation process, which might be challenging for practitioners without a background in psychometrics.
>
> **Weaknesses 3:** The integration of Item Response Theory (IRT) with machine learning evaluations assumes a level of prior knowledge (e.g., psychometric concepts) that could limit accessibility.
>
> **Answer 1 & 3:** Thank you for your thoughtful feedback and for highlighting the potential challenges of integrating Item Response Theory (IRT) and amortized calibration into machine learning evaluation frameworks, especially for practitioners less familiar with psychometric methods. We appreciate the opportunity to address these concerns and share our plans for making these methodologies more accessible and impactful.
>
> To bridge this gap, we aim to integrate IRT into widely-used generative model evaluation frameworks such as HELM and LM-eval-hardness. For example, the computerized adaptive test application of IRT demonstrated in our paper could be implemented as a built-in function within a dataloader in HELM. This integration, along with comprehensive documentation, will be made publicly available to support ease of use and reproducibility. We welcome suggestions for other evaluation frameworks where this methodology could add value and would greatly appreciate any recommendations from the reviewer community.
>
> Recognizing the importance of accessibility and reproducibility, we are prioritizing a user-friendly and well-documented implementation. To support further exploration and development, we have made our resources openly available on our GitHub repository: [Anonymous Repository](https://anonymous.4open.science/r/reeval). This repository aims to serve as a foundation for extending evaluation methodologies across machine learning, NLP, and related domains.
>
> Finally, while IRT may be new to some in the ML community, it is firmly rooted in latent variable modeling—a foundational topic covered in many standard machine learning textbooks. For example, Chapter 12.2.4 of *Pattern Recognition and Machine Learning* provides an excellent overview and is accessible [here](https://www.microsoft.com/en-us/research/uploads/prod/2006/01/Bishop-Pattern-Recognition-and-Machine-Learning-2006.pdf). We hope this context encourages its adoption and sparks further interest in its potential applications.
>
> Thank you again for your questions and suggestions, which have been invaluable in refining our approach and thinking about broader community engagement.
>
> **Weaknesses 2:** Details on the practical implementation of the item generator and its integration into existing workflows are limited.
>
> **Answer 2:**  Thank you for bringing up this important point. In response, we have revised Section 3 to provide additional details on the practical implementation of the item generator and its integration into existing workflows. Specifically, the item generator addresses gaps in the item bank by creating new items when no existing ones meet the desired difficulty level.  For example, if a target difficulty level $z$ is specified and no item in the bank aligns closely with it, the generator produces a new question tailored to the specified difficulty. These newly generated questions are then validated—either through LM-as-a-judge mechanisms or by human evaluators—to ensure their quality and relevance. Once validated, the items are added to the bank for future use, seamlessly enhancing the evaluation workflow. We believe these clarifications highlight the practical utility of the item generator and its role in streamlining and improving evaluation processes. Thank you again for your thoughtful feedback—it has been instrumental in improving the clarity and completeness of our work.

---

> ### Author Response · Authors · 2024-11-25
> **Responses for weaknesses (continue)**
>
> **Weaknesses 4:** The experiments predominantly rely on synthetic setups or controlled environments. Real-world case studies or collaborations with industry practitioners would strengthen the empirical validation.
>
> **Answer 4:** Thank you for raising this concern regarding our experimental setup. Our experiments were conducted on 24 real-world datasets sourced from HELM and evaluated across various state-of-the-art LLMs, including GPT-4 and Llama 3.1. For instance, both the calibration analysis and the test-independent ability estimation were performed using these real-world datasets. The only semi-synthetic component in our study pertains to the computerized adaptive testing (CAT) experiment. This approach is commonly used in active learning research, as recruiting new test takers for every experiment is often impractical. We greatly value your feedback and would be delighted to hear any suggestions for additional real-world experiments that could further strengthen our analysis. Your input would be invaluable in ensuring our work has the greatest possible impact and relevance.
>
> Regarding industry collaborations, we see significant potential for impactful partnerships with HELM, a leading large-scale leaderboard for AI model evaluation. Currently, HELM utilizes Classical Test Theory (CTT) scores on subsets to manage evaluation costs. However, our findings suggest that this approach may lack reliability in certain scenarios.  Our proposed IRT-based method offers a more reliable alternative while preserving the efficiency of subset testing, making it a practical and scalable solution for large-scale evaluation efforts. By integrating IRT into frameworks like HELM, we aim to enhance the robustness and accuracy of model evaluations, ultimately benefiting the broader AI community. We sincerely appreciate the opportunity to address these points and to emphasize the practical applicability and impact of our research. Thank you for your thoughtful consideration and support!

---

> ### Author Response · Authors · 2024-11-25
> **Responses for questions**
>
> Regarding your questions, we address them below:
>
> **Question 1:** The paper mentions missing values in the response matrix. How are these handled during calibration and scoring to ensure accurate ability estimation?
>
> **Answer 1:** Thank you for your insightful question. During both calibration and scoring, missing values are masked out when computing the likelihood, ensuring that they do not influence the calculations. Additionally, to mitigate potential numerical issues in our experiments, we exclude questions from the response matrix where all models either answer correctly or incorrectly. This step helps maintain the reliability of ability estimation and ensures the robustness of the evaluation process. We hope this explanation addresses your question. Please let us know if further details would be helpful.
>
> **Question 2:** Is the IRT model effective for non-text data, such as images or multimodal datasets? What adjustments are necessary?
>
> **Answer 2:** Yes, the IRT model can be applied to non-text data, including images and multimodal data. Currently, the IRT model we are studying operates on binary responses, which makes it naturally suited for tasks where evaluation outcomes can be categorized as success (1) or failure (0). For example, for image data, we need a grading method to assess the generated images from the model, which is a tool that the community has already had (such as vision-language-model-as-judge or human evaluator). Although we study the setting of binary response, we want to point out that there are IRT models tailored to non-binary metrics or tasks requiring finer-grained assessments [1]. This flexibility allows the IRT framework to adapt to a wide range of datasets and evaluation scenarios, making it applicable to diverse modalities. Expanding the model to address these adjustments is a natural extension of our work, and we are eager to explore this direction in future research.
>
>
> **Question 3:** How does the automated question generation avoid introducing bias, ensuring fair and objective evaluations?
>
> **Answer 3:** Thank you for this insightful question. We would like to first underscore the primary motivation for incorporating an automated question generator. It plays a supplemental role in adaptive testing, where questions are selected iteratively based on a desired difficulty level to evaluate model performance. While the primary source of questions is the original item pool, there may be instances where questions at a specific difficulty level are exhausted. In such cases, the item generator is employed to create additional questions. Importantly, this generator constitutes only a minor component of the evaluation process, with the majority of questions still derived from the original pool, thereby limiting the potential bias introduced by model-generated questions.
>
> We also acknowledge that automated question generation could serve broader purposes, such as replacing overused questions, expanding datasets, or constructing entirely new datasets. In these contexts, the potential for bias exists. However, as evidenced by modern AI benchmarks such as AIR-Bench [2], HEx-PHI [3], AdvBench [4], and SaladBench [5], the use of AI in dataset development is becoming increasingly common. A widely adopted practice is to involve human reviewers in the process to mitigate potential biases.
>
> We will revise our paper to emphasize the necessity of a collaborative framework where human experts and the automated item generator work together to ensure fairness and objectivity. The item generator excels in leveraging embedding representations to create questions at specific difficulty levels, a task that may surpass human intuition in precision and consistency. However, human reviewers remain indispensable in identifying, refining, and mitigating any biases that may arise in AI-generated content. This collaboration allows us to leverage the complementary strengths of both parties.
>
> We deeply appreciate the opportunity to address this important consideration and remain committed to advancing robust, fair, and efficient evaluation frameworks through continuous improvement and ethical practices. Thank you once again for highlighting this critical aspect of our work.
>
> [1] Ostini, R., & Nering, M. L. (2006). Polytomous item response theory models (No. 144). Sage.
>
> [2] Zeng, Y., et al. (2024). AIR-Bench 2024: A safety benchmark based on risk categories from regulations and policies. arXiv preprint arXiv:2407.17436.
>
> [3] Qi, X., et al. (2023). Fine-tuning Aligned Language Models Compromises Safety, Even When Users Do Not Intend To!. In The Twelfth International Conference on Learning Representations.
>
> [4] Zou, A., et al. (2023). Universal and transferable adversarial attacks on aligned language models. arXiv preprint arXiv:2307.15043.
>
> [5] Li, L., et al. (2024). Salad-bench: A hierarchical and comprehensive safety benchmark for large language models. ACL 2024.

---

> ### Comment · Reviewer_tNxg · 2024-11-25
> **Reply to Authors' responses**
>
> Thanks for your responses, which address my previous concerns. I will update my score.

---

### Author Response · Authors · 2024-12-01
**Thank you, reviewers and chairs, for working with us!**

Dear Reviewers and Chairs,

Thank you so much for working with us in the past few weeks on revising the papers. The process has helped us improve our work by addressing critical areas highlighted for revision. First, we conducted comprehensive ablation studies to evaluate the robustness and flexibility of our approach across various settings. These include the use of different IRT models (Rasch, 2PL, and 3PL), where Rasch achieved the best balance between complexity and performance; different base models for the conditioned item generator (Llama and Mistral), with results showing consistency; and different embedding models, where no significant oscillation in performance was observed. These studies collectively confirm the reliability of our framework.

Second, we added two new experiments to strengthen our empirical validation. The first experiment shows that model performance on the conditionally generated questions and the original dataset highly align with each other, demonstrating the reliability of using the generated questions to elicit model ability. The second experiment demonstrates that IRT-based evaluation has higher generalizability across different subsets than CTT in unbiased sampling scenarios because IRT has the mechanism to learn from side information.

Finally, we addressed minor writing inconsistencies, improved the clarity of our formatting, expanded the related work section, added a multilingual benchmark to our experiment, and included a clear and comprehensive list of the models evaluated in our study. To resolve confusion raised by reviewers, we clarified specific methodological details, facilitating better understanding and higher-level discussion. With these revisions, we demonstrate the robustness, scalability, and practicality of our methods, thoroughly addressing all reviewer concerns and significantly enhancing the quality of our paper.

We appreciate the opportunity to engage with the reviewers to improve our work. Please let us know if you have any additional questions. We thank you for your time and effort to support us.

Kind regards,
Authors

---

### Meta-Review · Area_Chair_WRfZ · 2024-12-20

**Metareview:**

This submission presents a new evaluation framework for LLMs by using Item Response Theory. The idea is to separate model performance from the selection of test sets, making evaluation more reliable and efficient. The author(s) introduce two strategies: amortized calibration to lower the cost of estimating item parameters and an item generator powered by a large language model to automate question generation. Very extensive evaluation experiments on many benchmarks and many language models demonstrate that the proposed strategy is more reliable and resource-efficient than traditional methods.

This submission has received six ratings (8, 8, 6, 6, 6, 5). The authors made very detailed and comprehensive rebuttal, which managed to convince reviewers tNxg, gy8L, and V1AP to update their ratings to 6, 8, 6, respectively. Reviewers D8pS (8), c8mZ (6), 9Mrp (5) left their overall assessments unchanged. One the reviewers shared concerns about this work on 1) the practical value of the proposed strategy, and 2) most test sets still containing a significant number of examples, and 3) the question generator. This reviewer confirmed that the author rebuttal did not provide evidence for these concerns during discussion.

Despite the submission's positive final ratings, the valuable strategies proposed, and extensive experiments, it has to receive a clear rejection recommendation due to a violation of ICLR's double-blind review policy. Three reviewers pointed out that the revised PDF submitted during the rebuttal contained the full author list and affiliations, which breaches the policy. As a result, the paper would be rejected.

**Additional Comments On Reviewer Discussion:**

This submission has received six ratings (8, 8, 6, 6, 6, 5). The authors made very detailed and comprehensive rebuttal, which managed to convince reviewers tNxg, gy8L, and V1AP to update their ratings to 6, 8, 6, respectively. Reviewers D8pS (8), c8mZ (6), 9Mrp (5) left their overall assessments unchanged. One the reviewers shared concerns about this work on 1) the practical value of the proposed strategy, and 2) most test sets still containing a significant number of examples, and 3) the question generator. This reviewer confirmed that the author rebuttal did not provide evidence for these concerns during discussion.

---

### Decision · Program_Chairs · 2025-01-22

Reject